# The Association between Coffee and Tea Consumption at Midlife and Risk of Dementia Later in Life: The HUNT Study

**DOI:** 10.3390/nu15112469

**Published:** 2023-05-25

**Authors:** Denise Abbel, Bjørn Olav Åsvold, Marit Kolberg, Geir Selbæk, Raymond Noordam, Håvard Kjesbu Skjellegrind

**Affiliations:** 1HUNT Research Centre, Department of Public Health and Nursing, NTNU, Norwegian University of Science and Technology, 7600 Levanger, Norway; n.t.abbel@lumc.nl (D.A.); bjorn.o.asvold@ntnu.no (B.O.Å.); 2Leiden University Medical Center, Department of Internal Medicine, Section of Gerontology and Geriatrics, 2333 ZA Leiden, The Netherlands; r.noordam@lumc.nl; 3K.G. Jebsen Center for Genetic Epidemiology, Department of Public Health and Nursing, NTNU, Norwegian University of Science and Technology, 7491 Trondheim, Norway; 4Department of Endocrinology, Clinic of Medicine, St. Olavs Hospital, Trondheim University Hospital, 7030 Trondheim, Norway; 5Center for Oral Health Services and Research Mid-Norway (TkMidt), 7030 Trondheim, Norway; marko@tkmidt.no; 6Norwegian National Centre for Aging and Health, Vestfold Hospital Trust, 3103 Tønsberg, Norway; geir.selbaek@aldringoghelse.no; 7Department of Geriatric Medicine, Oslo University Hospital, 0424 Oslo, Norway; 8Faculty of Medicine, Institute of Clinical Medicine, University of Oslo, 0318 Oslo, Norway; 9General Practice Research Unit, Department of Public Health and Nursing, Norwegian University of Science and Technology (NTNU), 7491 Trondheim, Norway; 10Levanger Hospital, Nord-Trøndelag Hospital Trust, 7600 Levanger, Norway

**Keywords:** coffee consumption, tea consumption, mild cognitive impairment, dementia, Alzheimer’s disease, nutrition, epidemiology, apolipoprotein E

## Abstract

Background: Studies exploring the possible protective effect of coffee and tea consumption on dementia have shown inconsistent results so far. We aimed to investigate whether consumption of tea and different types of coffee at midlife are associated with dementia later in life and whether sex or ApoE4 influence such association. Methods: We included 7381 participants from the Norwegian HUNT Study. Self-reported questionnaires assessed daily consumption of coffee and tea at baseline. After 22 years, individuals 70 years or older were screened for cognitive impairment. Results: General coffee consumption and tea consumption was not associated with dementia risk. Compared to daily consumption of 0–1 cups of coffee, daily consumption of ≥8 cups of boiled coffee was associated with increased dementia risk in women (OR: 1.83, 95% CI: 1.10–3.04, *p*-value for trend = 0.03) and daily consumption of 4–5 cups of other types of coffee was associated with a decrease in dementia risk in men (OR: 0.48, 95% CI: 0.32–0.72, *p*-value for trend = 0.05). Furthermore, the association between boiled coffee and increased dementia risk was only found in ApoE4 non-carriers. Differences by sex or ApoE4 carrier status were not supported by strong statistical evidence for interaction. Tea consumption was not associated with dementia risk. Conclusion: type of coffee may play a role in the direction of the association between coffee-drinking habits and dementia later in life.

## 1. Introduction

Globally, around 57.4 million people are currently living with dementia, which will increase to an estimated 152.8 million by 2050 [1]. Alzheimer’s disease (AD), which accounts for approximately 60–70% of all dementia cases [2], is characterized by amyloid-β plaques, neurofibrillary tangles, and neuroinflammation. Moreover, mild cognitive impairment (MCI) is a possible prodromal condition of dementia, and some individuals with MCI will eventually progress to dementia.

No disease-modifying treatment for cognitive decline or dementia has been established yet. Prevention is, therefore, of great importance. A previous study [3] theorized that 40% of dementia cases could be prevented or delayed by avoiding several modifiable risk factors, including physical inactivity, alcohol, and social isolation. Diet could be another promising strategy for dementia prevention [4]. Coffee and tea are two of the most commonly consumed beverages worldwide and contain many bioactive plant compounds [5]. Previous studies have shown that regular caffeine intake, which is found in both coffee and tea, reduced neuroinflammation and amyloid-β levels and reversed cognitive impairment in animal models [6,7]. One of the mechanisms of action of caffeine may be the competitive antagonism of adenosine receptors [8]. Nevertheless, evidence from observational studies researching the association between coffee and tea consumption and dementia risk is conflicting. While some studies conclude that increased coffee and tea consumption separately or in combination may be associated with a lower dementia risk [9,10,11], other studies report no association [12]. Most studies had a follow-up time of less than ten years. Preclinical Alzheimer’s disease may influence dietary habits through loss of initiative and social retraction. As the pathologic processes leading to clinical AD may have a preclinical phase of 10–15 years [13], studies with a shorter follow-up period are at risk for reverse causation bias.

While most studies are focused on universal risk reduction for the general population, assessing risk factors, including sex and genetics, could possibly enable optimal risk reduction by offering highly tailored approaches for dementia prevention. The most well-known genetic risk factor for Alzheimer’s disease is the Apolipoprotein E ε4 (ApoE4) allele [14]. ApoE4 carriers are at increased risk for dementia, and the environment may play a role in the translation of this genetic risk into the disease. Thus, this gene possibly influences associations between prevention interventions and dementia risk [15].

Lastly, little is known about the different brewing methods in the association between coffee and dementia development [16]. A coffee filter retains the lipids cafestol and kahweol, and consumption of unfiltered coffee could raise low-density lipoprotein (LDL)-cholesterol blood levels. This may increase the risk of cardiovascular diseases (CVD) and, thus, dementia [17,18,19].

In the current study, we used data from the Trøndelag Health (HUNT) Study, a longitudinal cohort study with a mean follow-up period of 21.8 years [20], to examine whether the consumption of tea and different types of coffee at midlife are associated with long-term risk of AD, MCI, and all-cause dementia, and to also assess the potential impact of sex and ApoE4 carrier status on this association.

## 2. Materials and Methods

### 2.1. Study Population and Design

Since 1984, the HUNT Study, a population-based cohort study, has been conducted in four surveys. HUNT1 (1984–1986), HUNT2 (1995–1997), HUNT3 (2006–2008), and HUNT4 (2017–2019) invited all residents in the Nord-Trøndelag region of Norway aged over 20 years. As part of HUNT4, a specific study on aging was performed, HUNT4 70+. Individuals aged 70 years or older, both community-dwelling and living in long-term care institutions, were invited to participate. The current study included the participants of HUNT2 in 1995–1997, who subsequently participated in the HUNT4 70+ Study in 2017–2019.

### 2.2. Exposure Assessment

Coffee and tea consumption were assessed during HUNT2 using a self-reported questionnaire. All participants were asked to write down the number of cups of boiled coffee, other types of coffee, and tea they consumed daily.

### 2.3. Outcome Assessment

Cognitive status at HUNT4, 22 years after exposure assessment, was used as the outcome. Details of dementia and cognition ascertainment are published elsewhere [21]. In brief, dementia and MCI were diagnosed based on the Diagnostic and Statistical Manual of Mental Disorders, Fifth Edition (DSM-5) criteria [22]. A consensus process was applied, using all available relevant information and at least two research and clinical experts from a group of nine specialists in neurology, geriatrics, and/or old age psychiatry, who independently assessed the cognitive status. When dementia was diagnosed, subtypes were defined based on clinical symptoms and categorized into: AD, vascular dementia, Lewy body dementia (LBD), frontotemporal dementia (FTD), mixed dementia, other specified dementia, or unspecified dementia. The main outcome of this study was overall dementia risk. Due to the low sample sizes of dementia subtypes other than AD, secondary outcomes were limited to MCI and AD risk.

### 2.4. Covariates

The following covariates, collected at baseline at HUNT2, were included: sex, age, marital status, educational attainment, smoking, physical activity (PA), marital status, alcohol consumption, history of CVD, history of diabetes mellitus (DM), body mass index (BMI), and ApoE4 carrier status. A more detailed description of the data collection in HUNT2 has been published previously [23]. In brief, information on sex, age, educational attainment, marital status, smoking, alcohol, PA, myocardial infarction, stroke, and diabetes was obtained using self-reported questionnaires. PA was assessed by self-reporting on weekly hours of light and vigorous PA (none/less than 1 h/1–2 h/≥3 h). Based on the answers, we calculated the metabolic equivalent (MET) hours per week (h/week). Less than 1 h/week was considered as 0.5 h/week, 1–2 h/week as 1.5 h/week, and ≥3 h/week as 3.5 h/week. For light PA, the amount of PA hours was multiplied by 2.5 METs, and for vigorous PA, PA hours were multiplied by 7 METs. Additionally, MET-h/week were grouped into: ≤8.3 MET-h/week, 8.4–16.6 MET-h/week and >16.6 MET-h/week.

During HUNT2, clinical measurements determined weight and height and were used to calculate BMI (kg/m^2^). Venous blood was drawn, and DNA was extracted from whole blood and used for genotyping. ApoE4 carrier status was determined using the two single-nucleotide polymorphisms (SNPs) rs7412 and rs429358.

### 2.5. Statistical Analyses

All statistical analyses were performed in STATA, version 17.0, and figures were made in R, version 4.0.3. Baseline characteristics of the study participants were reported as mean (standard deviation) for continuous variables and number (percentage) for categorical variables. We used logistic regression models to estimate odds ratios (ORs) with 95% confidence intervals (95% CI) of all-cause dementia according to different amounts of coffee consumption (0–1/2–3/4–5/6–7/≥8 cups daily) and tea consumption (0/1/2/≥3 cups daily). An amount of 0–1 cups of coffee were grouped together due to few individuals with low and no coffee consumption. These main analyses were adjusted according to the following three models: model 1 included sex (female/male) and age (years); model 2 was further adjusted for marital status (unmarried/married/widow(er)/divorced or separated), educational attainment (primary school/high school/college or university), smoking (never/previous/current daily smoker), alcohol consumption (total units/week), PA (≤8.3/8.4–16.6/>16.6 MET h/week), and tea/coffee consumption; and model 3 was additionally adjusted for BMI (<25/25–29.9/30–34.9/≥35 kg/m^2^), history of CVD (none/history myocardial infarction and/or stroke), and diabetes (yes/no). The analyses to estimate the association between the different types of coffee and dementia risk were adjusted according to model 4, which consisted of model 2 and was additionally adjusted for the alternative type of coffee (0–1/2–3/4–5/6–7/≥8 cups daily). To examine the robustness of our findings on the type of coffee, we repeated the analyses after excluding participants that drank the alternative type of coffee. Tests for linear trends in the OR across categories of exposure were performed using coffee and tea consumption categories as continuous variables.

Furthermore, multinomial logistic regression analyses estimated the relative risk (RR) with 95% CI for MCI and AD according to tea and different types of coffee consumption, adjusted according to model 4. In these analyses, participants diagnosed with other types of dementia were excluded.

To assess potential interaction effects, subgroup analyses according to model 4 were carried out according to the two potential effect modifiers, sex (female/men) and ApoE4 carrier status (negative/positive), to assess the dementia risk by coffee and tea consumption. For the ApoE4 analyses, we excluded participants with missing information on ApoE4 carrier status. We used the likelihood ratio test (LRT) to calculate the *p*-value for interaction, contrasting maximally adjusted models with and without the cross-product term of the stratifying variable with coffee or tea consumption in the model. To account for survival bias when assessing possible sex differences, we additionally performed sensitivity analysis where we ran the model, excluding participants older than 80 years at HUNT4.

### 2.6. Ethics

This project was approved by the Regional Committee for Medical and Health Research Ethics in Norway (456138), and the HUNT Study was approved by the Norwegian Data Protection Authority. Participation was voluntary, and all participants provided written informed consent.

## 3. Results

### 3.1. Participants

In total, 8758 people participated in both HUNT2 and HUNT4 70+ Studies (Figure 1), with a mean follow-up time of 21.8 years. Participants with missing data on coffee consumption, tea consumption, dementia diagnosis, or covariates (*n* = 1377) were excluded, resulting in 7381 individuals available for main analyses. For the ApoE4 sub-analyses and the analyses on AD/MCI risk, boiled coffee, and other types of coffee, we respectively excluded participants with missing data on the ApoE4 carrier status (*n* = 368), participants with other causes of cognitive impairment apart from MCI and AD (*n* = 417), participants who drank other types of coffee (*n* = 4561), and participants who drank boiled coffee (*n* = 3368).

In the total study sample, 985 (13.4%) cases of dementia were identified during HUNT4 70+ (Table 1). The mean age of the participants was 55.9 years (SD 6.2) at baseline, and the sample contained 3432 (46.5%) men. In general, people who solely completed primary school tended to drink more cups of coffee compared to people with a college or university education. Moreover, smokers tended to drink more cups of coffee than nonsmokers, and participants who were less physically active (≤8.3 MET-h/week) tended to drink more cups of coffee and fewer cups of tea compared to physically active participants (>16.6 MET-h/week). There was an inverse relationship between coffee and tea intake.

### 3.2. Coffee and Tea Consumption and Dementia Risk

In the regression analysis only adjusted for age and sex, daily consumption of ≥8 cups of coffee was associated with an increased risk of developing dementia (Model 1: OR: 1.45; 95% CI 1.05–2.01, *p*-value for trend = 0.02) compared to daily consumption of 0–1 cup of coffee (Table 2). Upon further adjustment for confounders, the association between coffee and dementia attenuated (Model 2: OR: 1.11; 95% CI: 0.78–1.57, *p*-value for trend = 0.81). Results for adjustment models 2 and 3 were similar. We found no other associations between general coffee consumption and dementia risk, nor was tea consumption associated with dementia risk.

### 3.3. Type of Coffee and Dementia Risk

According to adjustment model 2, daily consumption of ≥6 cups of boiled coffee was associated with an increased risk of dementia (6–7 cups: OR: 1.38, 95% CI: 1.05–1.81, ≥8 cups: OR: 1.46, 95% CI: 1.08–1.96, *p*-value for trend <0.01), compared to consumption of 0–1 cups of boiled coffee a day (Table 3). After further adjusting for other types of coffee, this association disappeared (6–7 cups: OR: 1.19, 95% CI 0.86–1.66, ≥8 cups: OR: 1.26, 95% CI 0.88–1.80, *p*-value for trend = 0.17), and associations between other consumption levels of boiled coffee and dementia were not found. In adjustment models 2 and 4, only the daily consumption of 4–5 cups of other types of coffee was associated with reduced dementia risk compared to drinking 0–1 cups daily (model 2: OR: 0.67, 95% CI: 0.54–0.82, *p*-value for trend <0.01, model 4: OR: 0.71 95% CI 0.54–0.92, *p*-value for trend = 0.24). The results according to adjustment model 4 were similar to adjustment model 3 and the sensitivity analyses models where we excluded participants who drank the alternative type of coffee. Additionally, sensitivity analysis using no coffee consumption as a reference provided similar results.

### 3.4. Coffee, Tea, and the Risk of MCI and AD

After adjusting according to model 4, daily consumption of 4–7 cups of boiled coffee was associated with an increased risk of MCI, compared to 0–1 cups of boiled coffee (Table 4) (4–5 cups: RR: 1.41; 95% CI 1.16–1.70, 6–7 cups: RR: 1.29 95% CI 1.01–1.63, *p*-value for trend = 0.04). Daily consumption of other types of coffee was not associated with MCI risk. Additionally, ≥6 cups of boiled coffee daily was associated with an increased risk of AD compared to daily consumption of 0–1 cups of boiled coffee (6–7 cups: RR: 1.85 95% CI 1.22–2.81, ≥8 cups: RR:1.65 95% CI 1.03–2.53, *p*-value for trend <0.01), and other types of coffee were not associated with AD risk. Daily tea consumption was also not associated with the risk of MCI or AD (Appendix A: Table A1). Results for all analyses were similar to adjustment model 3.

### 3.5. Sex Differences

After adjusting according to model 4, consumption of ≥8 cups of boiled coffee daily was associated with an increase in dementia risk in women compared to drinking 0–1 cups of boiled coffee daily (OR: 1.83, 95% CI: 1.10–3.04, *p*-value for trend = 0.03), and boiled coffee was not associated with dementia risk in men (*p*-value for trend = 0.66). There was little statistical evidence that the association differed between women and men (*p*-value for interaction = 1.00) (Figure 2). Additionally, we found that daily consumption of 4–5 cups of other types of coffee was associated with lower dementia risk in men (OR: 0.48, 95% CI: 0.32–0.72, *p*-value for trend = 0.05), but other types of coffee were not associated with dementia risk in women (*p*-value for trend = 0.84). Again, there was little statistical evidence that the association differed between women and men (*p*-value for interaction = 0.41). Adjusting for ApoE4 carrier status did not change the results). Additionally, when limiting the participants’ age at outcome measurement from 70 to 80 years, the sex differences in the direction of the association between coffee and dementia risk remained (Appendix A: Table A2). Lastly, in women and men, tea was not associated with dementia risk, and there was little statistical evidence that the association differed between women and men (*p*-value for interaction = 0.45) (Appendix A: Table A3).

### 3.6. ApoE4 Carrier Status

In ApoE4 non-carriers, 6–7 cups of boiled coffee were associated with increased dementia risk, compared to consumption of 0–1 cups of boiled coffee (OR: 1.53 95% CI: 1.00–2.32, *p*-value for trend = 0.10. This association was not found in ApoE4 carriers (OR: 1.02 95% CI: 0.47–1.48. *p*-value for trend = 0.57) (Table 5). There was little statistical evidence that the association differed by ApoE4 carrier status (*p*-value for interaction = 0.30). Other types of coffee were not associated with dementia risk in either ApoE4 subgroups (*p*-value for interaction = 0.55). Lastly, in both ApoE4 subgroups, tea was not associated with dementia risk (*p*-value for interaction = 0.92) (Appendix A: Table A4).

## 4. Discussion

This study, following 7381 participants with a mean follow-up time of 21.8 years, investigated the possible association between coffee and tea consumption and dementia risk. The main findings were as follows: (1) after adjusting for age and sex, general coffee consumption was associated with an increased dementia risk; however, upon further adjustment for confounders, the association disappeared. (2) Consumption of 4–7 cups of boiled coffee a day were associated with an increased risk of MCI, and ≥6 cups of boiled coffee daily was associated with an increased risk of AD. Consumption of other types of coffee was not associated with MCI and AD risk. (3) Daily consumption of 4–5 cups of other types of coffee was associated with reduced dementia risk in men but not in women. The daily consumption of ≥8 cups of boiled coffee was associated with an increased dementia risk in women but not in men. Daily consumption of 6–7 cups of boiled coffee was associated with an increased dementia risk in ApoE4 non-carriers but not in ApoE4 carriers. These suggestive differences by sex and ApoE4 carrier status were, however, not substantiated by statistical evidence for interaction. (4) Tea was not associated with dementia, MCI, or AD risk.

Previous studies showed inconsistent results concerning coffee consumption and dementia risk. A cohort study that followed 13,757 participants for eight years found that participants who had a daily intake of 2–2.9 cups and >3 cups of coffee had the lowest HRs for dementia incidence (0.69, 95% CI: 0.48–0.98 and 0.53, 95% CI: 0.31–0.89), compared to participants that consumed 0 cups of coffee daily [10]. On the other hand, a study including 3498 men with a follow-up period of 25 years found no significant association between coffee intake during midlife on the risk of cognitive impairment and dementia [12]. In addition, two studies reported only short-term associations between coffee intake and cognition, and no long-term effects were observed [24,25]. A possible explanation for the inconsistent results in previous studies could be the lack of differentiation between the types of coffee. Our study distinguished boiled and other types of coffee from the association between coffee and dementia risk. We showed that, in general, boiled coffee seems to be associated with an increased risk of developing dementia, while consumption of other types of coffee may be associated with a decrease in dementia risk. Although our study lacks information on the specifics of the other types of coffee, we assume that the other types of coffee mainly consisted of filtered coffee. Filtered coffee has been associated with lower cardiovascular risks compared to non-coffee drinkers and boiled coffee drinkers [26]. Boiled coffee is made by boiling or infusing course ground coffee for 5–10 min and is not filtered through a paper filter. It contains 30–50 times the amount of the coffee lipids cafestol and kahweol compared to filtered coffee [27] and increases the low-density lipoprotein LDL-cholesterol and, thus, the risk of atherosclerosis and cardiovascular diseases (CVDs) [17,28]. CVDs are associated with an increased risk of future dementia [18,19]. Although boiled coffee is not widely used anymore, current popular methods, including French press, Turkish, and Greek coffee, have comparable amounts of cafestol and kahweol due to the absence of a paper filter [27]. Espresso contains intermediate amounts of cafestol and kahweol. Although causality cannot be established, this study highlights that coffee might affect dementia risk and may, therefore, be used as a cost-effective complement to prevention strategies.

The results of this current study suggest that women had an increased dementia risk when drinking higher amounts of boiled coffee, while men had a dementia risk reduction when drinking other types of coffee. The direction of the associations and significance remained after excluding participants aged over 80 years at HUNT4. These suggestive differences were, however, not substantiated by statistical evidence for interaction. The previous literature showed conflicting findings on sex differences in the association between coffee and dementia risk. While some studies showed that coffee consumption was more strongly associated with a reduction in dementia risk in men [10], others showed that coffee consumption was especially associated with a dementia risk reduction in women [9,29]. Although evidence favoring possible causality cannot be established through the observational nature of our study, we can speculate about the possible mechanisms behind the possible sex differences in the association between coffee and dementia risk. The antioxidant effect of coffee might be more beneficial in men than in women. Both human and animal studies have reported that oxidative stress levels were higher in men than in women [30,31], and Ishizaka, Yamakado [32] found that coffee consumption, in a graded matter, was associated with decreased oxidative stress levels in men but not in women. If men truly benefit more from coffee consumption, this may balance out the negative effects of boiled coffee and could explain why we found that boiled coffee was associated with an increased dementia risk in women but not in men. More research is needed to determine the sex differences in coffee consumption and the possible influence of this on dementia development.

Moreover, our results showed that in ApoE4 non-carriers, boiled coffee was associated with an increased dementia risk, while this association was not found in ApoE4 carriers. These suggestive differences were, however, not substantiated by statistical evidence for interaction. ApoE4 is an apolipoprotein that plays a major role in the transport of cholesterol, as it is a LDL-receptor ligand [33]. A population-based cohort study found that increased levels of LDL-cholesterol were associated with increased risk of AD in ApoE4 non-carriers but not in ApoE4 carriers [34], and another cohort study showed that higher LDL-cholesterol was associated with decreased verbal memory in ApoE4 non-carriers but not in ApoE4 carriers [35]. One could speculate that ApoE4 non-carriers may be more at risk for dementia when drinking the lipids that are found in boiled coffee, but the precise underlying mechanisms remain unknown.

We found no association between tea consumption and MCI, AD, and dementia risk. However, tea drinking was relatively rare in our population and led to decreased statistical power. In line with our results, the Finnish Cardiovascular Risk Factor, Aging, and Dementia (CAIDE) study reported that only coffee, but not tea, was associated with dementia risk [36]. However, another study, with a follow-up period of 11.4 years, concluded that tea, separately or in combination with coffee, was associated with a decrease in dementia risk [11]. Compared to non-drinkers, 2–3 cups of coffee or tea were associated with 28% lower dementia. The type and amount of possible beneficial components differ per tea type [37]. In our study, there was no information available on the type of tea. Black tea dominates the market in Norway [38], and it is thus plausible that the majority of our tea-drinking participants drank black tea. Several studies showed that green tea was associated with a reduced dementia risk, while black tea was not [39,40].

The precise protective mechanisms of non-boiled coffee against dementia are unknown. The antagonist binding of caffeine to adenosine receptors, which stimulates cholinergic neurons (Fredholm et al., 1999), is a possible pathway that may explain the association. Through the competitive antagonist binding of adenosine receptors, caffeine reduced neuroinflammation and neurodegeneration in rats [7,41]. Moreover, the antioxidants in coffee might influence the risk of dementia by reducing oxidative stress [42]. Preclinical studies indicate the neuroprotective potential of several polyphenolic compounds found in coffee [43]. Lastly, coffee consumption has been associated with reduced risk of several diseases, including Parkinson’s disease [44,45], diabetes [46,47], and cardiovascular disease [26,48]. The previously mentioned comorbidities are risk factors for dementia, and decreasing the risk for comorbidities may mediate the association between coffee and tea and dementia risk. In our study, the associations did not substantially change after additional adjustment for comorbidities. However, as the participants had a mean age of 55.9 (SD = 6.3) at baseline, the prevalence of comorbidities at baseline was low.

Our study has several strengths. Firstly, while most previous epidemiological studies had shorter follow-up periods, this current study has a mean follow-up period of 21.8 years, making our study less prone to reverse causation bias. Other strengths of this current study are the participation of both home-dwelling and institutionalized elderly people, controls without dementia who were identified through diagnostic tests and ability to adjust for lots of confounders. Several limitations should also be taken into account when interpreting our results. Firstly, when assessing coffee consumption, the question remains what the alternative is. One must note that all participants drank something. Coffee may be more or less harmful than other drinks and is also related to other dietary habits. Therefore, our analyses may not completely reflect the true association. Cognitive assessment only at the last follow-up at HUNT4 may be another limitation. Additionally, covariates and consumption of coffee and tea were surveyed at baseline and may not reflect long-term drinking habits. Lifestyle factors are dynamic. For example, during the follow-up period, most participants retired, which may lead to major changes in daily habits. Moreover, our results have wide CIs, representing uncertainty about the effect size. Additionally, we had limited statistical power to assess interactions. Although our participation rates were relatively high, non-participants had lower socioeconomic status and higher mortality than individuals that participated in the HUNT Study [20,49]. This may limit the generalizability of our findings. Moreover, this study population was predominantly white, so our results may be less representative for populations with other ethnic backgrounds. Additionally, although a systematic approach was used to diagnose people with dementia, we did not have access to biomarker data for the diagnosis of dementia and AD. Lastly, when researching older individuals aged over 70 years, the chance of competing risk from death and participation bias based on cognitive status are unavoidable constraints.

To conclude, consumption of 4–7 cups of boiled coffee a day was associated with an increased risk of MCI, and ≥6 cups of boiled coffee daily was associated with an increased risk of AD. Consumption of other types of coffee was not associated with MCI or AD risk. Moreover, boiled coffee was associated with increased dementia risk in women and ApoE4 non-carriers, and consumption of other types of coffee was associated with decreased dementia risk in men. Tea consumption was not associated with dementia risk. These suggestive differences in sex and ApoE4 carrier status were, however, not substantiated by statistical evidence for interaction. Since coffee and tea are among the most widely consumed beverages worldwide and cognitive decline is common among older adults, even minor potential health benefits or risks associated with coffee and tea consumption could have a significant effect on public health. Long-term intervention studies are needed to disentangle the possible relationship between coffee intake, dementia risk, type of coffee, sex, and ApoE4 carrier status.

## Figures and Tables

**Figure 1 nutrients-15-02469-f001:**
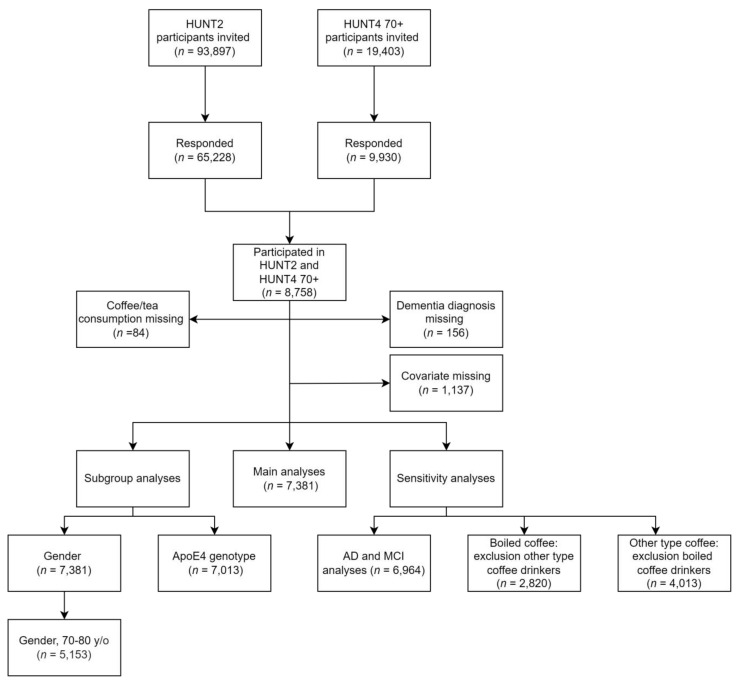
Flowchart of participant selection.

**Figure 2 nutrients-15-02469-f002:**
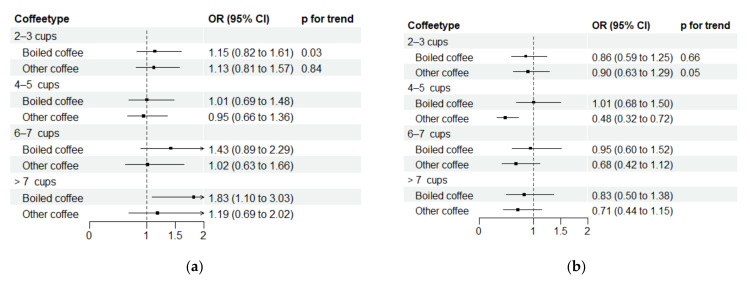
OR (95% CI) of dementia risk by type of coffee consumption and sex, adjusted according to model 4. (**a**) Women; (**b**) men. Compared to the 0–1 cup/day category. Results were adjusted for participation age (years), sex (female/men), educational attainment (primary school/high school/college or university), marital status (unmarried/married/widow(er)/divorced or separated), smoking status (never/previous/currently daily smoker), tea consumption (0/1/2/≥3), exercise (MET-h/week score), alcohol consumption (total units/week), and the alternative coffee type consumption (0–1/2–3/4–5/6–7/≥8).

**Table 1 nutrients-15-02469-t001:** Characteristics of study sample examining the possible association between coffee and dementia risk, collected at HUNT2 and HUNT4.

		Coffee Consumption (Cups/Day)					Tea Consumption (Cups/Day)			
	Overall	0–1	2–3	4–5	6–7	≥8	0	1	2	≥3
Total study population, n	7381	580	1770	2584	1248	1199	4413	1132	1184	652
Sex, male	3432 (46.5)	286 (49.3)	768 (43.4)	1099 (42.5)	594 (47.6)	685 (57.1)	2229 (50.5)	497 (43.9)	411 (34.7)	295 (45.2)
Age	55.85 ± 6.20	56.22 ± 6.19	56.57 ± 6.56	56.18 ± 6.43	55.40 ± 5.96	54.39 ± 5.36	55.76 ± 6.21	55.83 ± 6.31	56.22 ± 6.47	55.90 ± 5.96
Educational attainment										
Primary school	2976 (40.3)	162 (27.9)	633 (35.8)	1104 (42.7)	532 (42.6)	545 (45.5)	1918 (43.5)	363 (32.1)	454 (38.3)	241 (37.0)
High school	2655 (36.0)	222 (38.3)	642 (36.3)	891 (34.5)	452 (36.2)	448 (37.4)	1578 (35.8)	425 (37.5)	432 (36.5)	220 (33.7)
College/university	1750 (23.7)	196 (33.8)	495 (28.0)	589 (22.8)	264 (21.2)	206 (17.2)	917 (20.8)	344 (30.4)	298 (25.2)	191 (29.3)
Marital status										
Unmarried	310 (4.2)	42 (7.2)	70 (4.0)	104 (4.0)	47 (3.8)	47 (3.9)	197 (4.5)	42 (3.7)	41 (3.5)	30 (4.6)
Married	6097 (82.6)	462 (79.7)	1457 (82.3)	2137 (82.7)	1057 (84.7)	984 (82.1)	3635 (82.4)	945 (83.5)	984 (83.1)	533 (81.7)
Widow(er)/divorced/separated	974 (13.2)	76 (13.1)	243 (13.7)	343 (13.3)	144 (11.5)	168 (14.0)	581 (13.2)	145 (12.8)	159 (13.4)	89 (13.7)
Tea consumption (cups/day)										
0	4413 (59.8)	181 (31.2)	721 (40.7)	1595 (61.7)	935 (74.9)	981 (81.8)				
1	1132 (15.3)	103 (17.8)	436 (24.6)	398 (15.4)	120 (9.6)	75 (6.3)				
2	1184 (16.0)	119 (20.5)	397 (22.4)	438 (17.0)	139 (11.1)	91 (7.6)				
≥3	652 (8.8)	177 (30.5)	216 (12.2)	153 (5.9)	54 (4.3)	52 (4.3)				
Coffee consumption (cups/day)										
0–1	580 (7.9)						181 (4.1)	103 (9.1)	119 (10.1)	177 (27.1)
2–3	1770 (24.0)						721 (16.3)	436 (38.5)	397 (33.5)	216 (33.1)
4–5	2584 (35.0)						1595 (36.1)	398 (35.2)	438 (37.0)	153 (23.5)
6–7	1248 (16.9)						935 (21.2)	120 (10.6)	139 (11.7)	54 (8.3)
≥8	1199 (16.2)						981 (22.2)	75 (6.6)	91 (7.7)	52 (8.0)
BMI (kg/m2)										
<25	2449 (33.2)	223 (38.4)	613 (34.6)	853 (33.0)	383 (30.7)	377 (31.4)	1392 (31.5)	407 (36.0)	403 (34.0)	247 (37.9)
25–29.9	3759 (51.0)	272 (46.9)	888 (50.2)	1317 (51.0)	671 (53.8)	611 (51.0)	2286 (51.8)	575 (50.8)	597 (50.4)	301 (46.2)
30–34.9	939 (12.7)	61 (10.5)	210 (11.9)	333 (12.9)	163 (13.1)	172 (14.3)	597 (13.5)	113 (10.0)	146 (12.3)	83 (12.7)
≥35	234 (3.2)	24 (4.1)	59 (3.3)	81 (3.1)	31 (2.5)	39 (3.3)	138 (3.1)	37 (3.3)	38 (3.2)	21 (3.2)
Alcohol (units/week)	1.71 ± 2.22	1.27 ± 2.13	1.61 ± 2.19	1.68 ± 2.13	1.78 ± 2.07	2.06 ± 2.58	1.78 ± 2.36	1.68 ± 2.02	1.50 ± 1.94	1.65 ± 2.12
PA (MET-h/week)										
≤8.3	3930 (53.2)	303 (52.2)	908 (51.3)	1363 (52.7)	672 (53.8)	684 (57.0)	2439 (55.3)	558 (49.3)	618 (52.2)	315 (48.3)
8.3–16.6	2273 (30.8)	168 (29.0)	575 (32.5)	812 (31.4)	393 (31.5)	325 (27.1)	1302 (29.5)	383 (33.8)	369 (31.2)	219 (33.6)
>16.6	1178 (16.0)	109 (18.8)	287 (16.2)	409 (15.8)	183 (14.7)	190 (15.8)	672 (15.2)	191 (16.9)	197 (16.6)	118 (18.1)
Smoking										
Never	3305 (44.8)	402 (69.3)	1028 (58.1)	1186 (45.9)	441 (35.3)	248 (20.7)	1682 (38.1)	636 (56.2)	638 (53.9)	349 (53.5)
Previous	2579 (34.9)	133 (22.9)	571 (32.3)	943 (36.5)	481 (38.5)	451 (37.6)	1608 (36.4)	351 (31.0)	395 (33.4)	225 (34.5)
Current	1497 (20.3)	45 (7.8)	171 (9.7)	455 (17.6)	326 (26.1)	500 (41.7)	1123 (25.4)	145 (12.8)	151 (12.8)	78 (12.0)
DM, yes	141 (1.9)	14 (2.4)	32 (1.8)	49 (1.9)	29 (2.3)	17 (1.4)	69 (1.6)	25 (2.2)	31 (2.6)	16 (2.5)
CVD, at least one	179 (2.4)	11 (1.9)	43 (2.4)	59 (2.3)	26 (2.1)	40 (3.3)	120 (2.7)	25 (2.2)	20 (1.7)	14 (2.1)
ApoE4 carrier status, positive	2101 (30.0)	182 (33.1)	480 (28.7)	721 (29.4)	368 (31.1)	350 (30.3)	1255 (29.8)	315 (29.1)	348 (31.2)	183 (30.0)
Cognitive status										
No CI	3840 (55.1)	331 (61.1)	927 (55.8)	1344 (55.1)	636 (53.6)	602 (53.0)	2230 (53.7)	636 (59.0)	617 (55.2)	357 (58.1)
MCI	2552 (36.7)	177 (32.7)	584 (35.2)	896 (36.7)	448 (37.8)	447 (39.4)	1571 (37.8)	358 (33.2)	412 (36.9)	211 (34.4)
Dementia, all causes	985 (13.4)	71 (12.2)	259 (14.6)	343 (13.3)	163 (13.1)	149 (12.4)	610 (13.8)	136 (12.0)	155 (13.1)	84 (12.9)
AD	572 (8.2)	34 (6.3)	150 (9.0)	200 (8.2)	102 (8.6)	86 (7.6)	354 (8.5)	84 (7.8)	88 (7.9)	46 (7.5)

Values are expressed as mean ± standard deviation or number (%). Abbreviations: AD = Alzheimer’s disease; BMI = body mass index; CI = cognitive impairment; CVD = cardiovascular diseases; DM = diabetes mellitus; MCI = mild cognitive impairment; PA = physical activity.

**Table 2 nutrients-15-02469-t002:** OR (95% CI) of dementia by consumption of coffee and tea, stepwise adjusted.

	Coffee Consumption (Cups/Day)						Tea Consumption (Cups/Day)				
	0–1	2–3	4–5	6–7	≥8	*p*-Value for Trend	0	1	2	≥3	*p*-Value for Trend
Model 1	Ref	1.16 (0.86–1.57)	1.09 (0.81–1.46)	1.26 (0.91–1.73)	1.45 (1.05–2.01)	0.02	Ref	0.81 (0.66–1.01)	0.84 (0.69–1.03)	0.91 (0.70–1.18)	0.11
Model 2	Ref	1.12 (0.82–1.52)	0.95 (0.70–1.30)	1.06 (0.75–1.48)	1.11 (0.78–1.57)	0.81	Ref	0.92 (0.74–1.15)	0.92 (0.75–1.14)	1.02 (0.77–1.34)	0.71
Model 3	Ref	1.11 (0.81–1.51)	0.94 (0.69–1.28)	1.04 (0.74–1.46)	1.09 (0.77–1.54)	0.90	Ref	0.92 (0.74–1.16)	0.92 (0.74–1.14)	1.01 (0.76–1.33)	0.65

Model 1: results were adjusted for participation age (years), sex (female/men). Model 2: results were additionally adjusted for educational attainment (primary school/high school/college or university), marital status (unmarried/married/widow(er)/divorced or separated), smoking status (never/previous/currently daily smoker), coffee/tea consumption, exercise (MET-h/week score), and alcohol consumption (total units/week). Model 3: results were further adjusted for diabetes (yes/no), CVD (none/at least one), and BMI (<25, 25–29.9, 30–34.9, ≥35).

**Table 3 nutrients-15-02469-t003:** OR (95% CI) of dementia risk by consumption of boiled and other types of coffee consumption, adjusted according to models 2 and 4.

Coffee Consumption (Cups/Day)							
		0–1	2–3	4–5	6–7	≥8	*p*-Value for Trend
Boiled coffee	*n*	4284	1123	1006	511	457	
	Model 2	Ref	1.14 (0.93–1.41)	1.14 (0.92–1.41)	1.38 (1.05–1.81)	1.46 (1.08–1.96)	<0.01
	Model 4	Ref	1.01 (0.79–1.30)	1.00 (0.75–1.30)	1.19 (0.86–1.66)	1.26 (0.88–1.80)	0.17
Other types of coffee	*n*	3148	1447	1540	639	607	
	Model 2	Ref	0.96 (0.79–1.18)	0.67 (0.54–0.82)	0.80 (0.60–1.07)	0.86 (0.64–1.15)	<0.01
	Model 4	Ref	1.01 (0.80–1.28)	0.71 (0.54–0.92)	0.86 (0.61–1.21)	0.93 (0.65–1.32)	0.24

Model 2: results were adjusted for participation age (years), sex (female/men), educational attainment (primary school/high school/college or university), marital status (unmarried/married/widow(er)/divorced or separated), smoking status (never/previous/currently daily smoker), tea consumption (0/1/2/≥3), exercise (MET–h/week score), and alcohol consumption (total units/week). Model 4: results were adjusted according to model 2 and additionally adjusted for the alternative coffee type consumption (0–1/2–3/4–5/6–7/≥8).

**Table 4 nutrients-15-02469-t004:** RR (95% CI) of MCI and AD risk by coffee consumption, adjusted according to model 4.

Coffee Consumption (Cups/Day)								
			0–1	2–3	4–5	6–7	≥8	*p*-Value for Trend
MCI	Boiled coffee	*n*	1405	369	407	199	172	
			Ref	1.03 (0.87–1.22)	1.41 (1.16–1.70)	1.29 (1.01–1.63)	1.11 (0.86–1.43)	0.04
	Other types of coffee	*n*	1138	462	521	212	219	
			Ref	0.98 (0.83–1.16)	0.97 (0.81–1.15)	0.94 (0.75–1.18)	1.01 (0.80–1.28)	0.87
AD	Boiled coffee	*n*	279	94	97	59	43	
			Ref	1.15 (0.83–1.58)	1.37 (0.96–1.95)	1.85 (1.22–2.81)	1.65 (1.03–2.53)	<0.01
	Other types of coffee	*n*	295	109	92	41	35	
			Ref	1.14 (0.83–1.55)	0.79 (0.56–1.12)	1.00 (0.64–1.56)	1.03 (0.64–1.67)	0.71

Results were adjusted for participation age (years), sex (female/men), educational attainment (primary school/high school/college or university), marital status (unmarried/married/widow(er)/divorced or separated), smoking status (never/previous/currently daily smoker), tea consumption (0/1/2/≥3), exercise (MET-h/week score), alcohol consumption (total units/week), and the alternative coffee type consumption (0–1/2–3/4–5/6–7/≥8).

**Table 5 nutrients-15-02469-t005:** OR (95% CI) of AD risk by type of coffee consumption and ApoE4 carrier status, adjusted according to model 4.

Coffee Consumption (Cups/Day)								
	ApoE4		0–1	2–3	4–5	6–7	≥8	*p*-Value for Trend
Boiled coffee	Non-carrier	*n*	2844	753	675	330	310	
			Ref	1.15 (0.84–1.58)	1.15 (0.81–1.64)	1.53 (1.00 –2.32)	1.31 (0.82–2.09)	0.10
	Carrier	*n*	1227	314	281	156	123	
			Ref	0.87 (0.56–1.35)	0.80 (0.44–1.46)	1.02 (0.47–1.48)	1.45 (0.79–2.66)	0.57
Other types of coffee	Non-carrier	*n*	2070	983	1031	417	411	
			Ref	1.14 (0.84–1.55)	0.77 (0.54–1.09)	0.81 (0.51–1.29)	0.86 (0.54–1.39)	0.19
	Carrier	*n*	912	392	433	29	181	
			Ref	0.78 (0.51–1.20)	0.61 (0.39–0.96)	0.85 (0.48–1.52)	1.05 (0.58–1.89)	0.84

Results were adjusted for participation age (years), sex (female/men), educational attainment (primary school/high school/college or university), marital status (unmarried/married/widow(er)/divorced or separated), smoking status (never/previous/currently daily smoker), tea consumption (0/1/2/≥3), exercise (MET-h/week score), alcohol consumption (total units/week), and the alternative coffee type consumption (0–1/2–3/4–5/6–7/≥8).

## Data Availability

Restrictions apply to the availability of these data. Data were obtained from HUNT Databank. HUNT data are available for research projects approved by a Norwegian Regional Committee for Medical Research Ethics (REK).

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
