# Peer review of "The Association between Coffee and Tea Consumption at Midlife and Risk of Dementia Later in Life: The HUNT Study"

_nutrients, 2023, doi:10.3390/nu15112469_

Round 1

Reviewer 1 Report

This article aims to investigate the association between coffee and tea consumption and dementia in adults. The study is well-structured, however, some aspects should be better specified.

Minor comments:

  • In general please revise the editing, i.e., please check the editing of the “Participants” paragraph; please use one style to report OR in the text (OR: or OR=), check the caption of Table 2 should be “of coffee and tea”, editing of Figure 2 instead of “coffeetype” should be “coffee type”;

  • Table 5 please explain to what refers “positive” and “negative”; better to specify “ApoE4 carriers” and “ApoE4 non-carriers”.

  • Line 31 abstract, please specify the direction of the association;

  • Several of the significant results were not reported in the abstract and discussed in the discussion section, please check;

  • I advise the authors to include a paragraph in the discussion section, with an outline of the recent evidence on the possible mediating effect of tea and coffee polyphenols in cognitive health. Please underline the presence and the influence of coffee and tea polyphenols in the brain, especially some polyphenols which are able to cross the blood-brain barrier (BBB) (i.e. caffeic acid) (PMID: 32466115). I suggest focusing on the potential neuroprotective mechanisms exerted by coffee polyphenols (PMID: 34624428).

Reviewer 2 Report

Review comments for manuscript ID nutrients-2340715

Brief summary

This manuscript outlines the associations between self-reported coffee/tea intakes and dementia developed later in life in a cohort of mid-life adults and followed-up in time over 22 years, in a relatively large sample size (n=7381). The authors also verified their results with sensitivity analyses to account for a range of potential confounding factors. One of the main strengths in this study is the availability of ApoE4 data that the authors incorporated into their analysis, and found the association between coffee intake and dementia risk was found only in ApoE4 non-carriers. Another strength of this study is that participants were follow-up in time, which enables the authors to investigate the follow-up coffee/tea intakes and follow-up dementia diagnosis, which add further value to previous coffee-association epidemiological studies.

General concept comments

However, there are some comments for the editor and authors’ consideration for clarifying certain sections of the manuscript.

In its current form, it is unclear when the coffee/tea intakes and dementia diagnosis was performed at baseline or at the follow-up timepoint. Did the participants who were diagnosed with dementia at baseline also diagnosed with dementia at follow-up and how many? Is coffee/tea intake patterns similar at baseline and at follow-up? As there is longitudinal data available and referring to the title, could a cox proportional hazards regression analysis be more relevant? Or is the analysis actually focused on using baseline coffee/tea intake to investigate the association with dementia diagnosed at the follow-up timepoint? If so, then the title and methods should accurately reflect this. And if so, how would the authors justify the associations using coffee/tea intakes from 22 years prior to investigate the association with dementia developed 22 years later?

One of the main strengths of this work is the availability of ApoE4 data that the authors incorporated into their analysis, and found the association between coffee intake and dementia risk was found only in ApoE4 non-carriers. The authors also mentioned that all dementia subtypes were grouped together. Could the authors explain how relevant the two selected SNPs are to AD and other forms of dementia/cognitive decline?

Concerning dementia and cognitive assessment (see section 2.3 outcome assessment), it is not clear if MCI/dementia was diagnosed at baseline or at follow-up or at both timepoints. As another strength of this study is the follow-up timepoint, it would add value to the manuscript to describe if dementia diagnosis was performed at both timepoints, and if so, to describe/compare this statistics at baseline to the 985 cases 913.4%) described in results section 3.1. 

Concerning coffee and tea intakes, the authors described “boiled coffee and other types of coffee/tea consumed daily” (see section 2.2 exposure assessment). Is boiled coffee the main type of coffee consumed in Norway during the respective period and consumed more frequently? It would be clearer to provide a short rationale of why boiled coffee was emphasised and what other types of coffee/tea questions in the questionnaire were, and if additives to coffee/tea intake were also reported or considered for statistical analysis, especially as the authors reported that in section 3.3, that the association between boiled coffee and dementia risk disappeared after accounting for “other types of coffee”. The authors should also clarify if the coffee/tea intakes collected in the baseline questionnaire or available at both baseline and follow-up timepoints? How skewed is the distribution for coffee/tea intakes and was any transformation applied to the coffee/tea intake for downstream statistical analysis? Why is the categorisation for number of cups/day of tea and coffee different?

Specific comments

Abstract

Line 22: spelling mistake for putatitive

Results

Table 1: The category for coffee and tea intakes appear to be different, with non-tea drinkers categorised into a separate category from 1 cup/day drinkers- different to coffee 0-1 cup/day drinkers. Is this true or could be easily mis-interpretated. Please clarify/revise the table, unless this was an intentional categorisation, then the authors should justify this in the method section.

Section 3.1 or 3.6: The authors should also describe, amongst the participants classified with dementia/risk, how many/percent were ApoE4 non-carriers.

Line 185: how was this relationship assessed-statistically or visually?

Line 195: as mentioned above, please be clear if 0 and 1 cup of coffee drinkers were grouped into the same category. If they are, please provide the rationale, as non-coffee drinkers should be not grouped as coffee-drinkers, which is different to the non-tea and tea-drinkers categorisation applied in this same study. Revising the category for 0 cup/day of coffee may better reflect a more accurate association than 0-1 cup/day of coffee, please revise this for the other-related analyses.

Line 195: are the coffee intake and dementia diagnosis data collected at baseline or at follow-up timepoint? 

Line 317: I agree with the authors on this point and would suggest for the authors to include a supplementary table to show the proportion of boiled coffee to “other types of coffee”, categorised by cups of coffee/day.

Line 324: It is interesting point that the authors made. Could the authors clarify briefly in a sentence how boiled coffee is made? To enable the reader to understand the difference from filtered coffee, which is used for comparison in this study. Are very fine coffee grounds used such that a filter is not needed? 

Line 328: French press may not use a paper filter but has a mesh filter.

Lines 333: the authors should be cautious to make this generalised statement as this association for women in this study was only for >8 cups of coffee, specifically boiled coffee, and without adjusting for ApoE4 status (in the sex-specific analysis). 8 cups of coffee is an extremely high intake. How commonly/frequently are 8 cups of coffee consumed in the Norwegian population? 

Lines 344-350: was this trend observed for high coffee consumption?

Line 355: Were there sex differences in ApoE4 status?

Lines 362-364: This sentence is too suggestive. If the authors were to substantiate this comment, they would need to justify with references that show different lipid content/composition of boiled vs other types of coffee.
